# A locus conferring tolerance to *Theileria* infection in African cattle

**David Wragg**[1,2☉], **Elizabeth A. J. Cook**[3,4☉], **Perle Latré de Laté**[3,4], **Tatjana Sitt**[4], **Johanneke D. Hemmink**[1,2,3,4], **Maurine C. Chepkwony**[4], **Regina Njeru**[3,4], **E. Jane Poole**[4], **Jessica Powell**[2], **Edith A. Paxton**[2], **Rebecca Callaby**[2,5], **Andrea Talenti**[2], **Antoinette A. Miyunga**[3,4], **Gideon Ndambuki**[3,4], **Stephen Mwaura**[4], **Harriet Auty**[6], **Oswald Matika**[2], **Musa Hassan**[1,2], **Karen Marshall**[3,4], **Timothy Connelley**[1,2], **Liam J. Morrison**[1,2], **B. Mark deC. Bronsvoort**[1,2], **W. Ivan Morrison**[1,2], **Philip G. Toye**[3,4‡]*, **James G. D. Prendergast**[1,2‡]*

1 Centre for Tropical Livestock Genetics and Health (CTLGH), Easter Bush Campus, Edinburgh, United Kingdom, 2 The Roslin Institute, University of Edinburgh, Edinburgh, United Kingdom, 3 Centre for Tropical Livestock Genetics and Health (CTLGH), ILRI Kenya, Nairobi, Kenya, 4 ILRI Kenya, Nairobi, Kenya, 5 The Epidemiology, Economics and Risk Assessment (EEA) Group, Easter Bush Campus, Edinburgh, United Kingdom, 6 Institute of Biodiversity Animal Health & Comparative Medicine, College of Medical, Veterinary & Life Sciences, University of Glasgow, Glasgow, United Kingdom

☉ These authors contributed equally to this work.
‡ PGT and JGDP also contributed equally to this work.
* p.toye@cgiar.org (PGT); James.Prendergast@ed.ac.uk (JGDP)

**Data Availability Statement:** Whole-genome sequencing data generated from the field trial animals is available to download from the European Nucleotide Archive (ENA) under project accession PRJEB39210. The publicly available

## Abstract

East Coast fever, a tick-borne cattle disease caused by the *Theileria parva* parasite, is among the biggest natural killers of cattle in East Africa, leading to over 1 million deaths annually. Here we report on the genetic analysis of a cohort of *Bos indicus* (Boran) cattle demonstrating heritable tolerance to infection with *T. parva* ($h^2$ = 0.65, s.e. 0.57). Through a linkage analysis we identify a 6 Mb genomic region on bovine chromosome 15 that is significantly associated with survival outcome following *T. parva* exposure. Testing this locus in an independent cohort of animals replicates this association with survival following *T. parva* infection. A stop gained variant in a paralogue of the *FAF1* gene in this region was found to be highly associated with survival across both related and unrelated animals, with only one of the 20 homozygote carriers (T/T) of this change succumbing to the disease in contrast to 44 out of 97 animals homozygote for the reference allele (C/C). Consequently, we present a genetic locus linked to tolerance of one of Africa's most important cattle diseases, raising the promise of marker-assisted selection for cattle that are less susceptible to infection by *T. parva*.

## Author summary

More than a million cattle die of East Coast fever in Africa each year, the impact of which disproportionately falls onto low-income, smallholder farmers. The lack of a widely accessible vaccine, heavy reliance on chemicals to control the tick vector and inadequate drug treatments means that new approaches for controlling the disease are urgently required.

WGS data is available from the ENA under project accessions: PRJEB12739, PRJEB14552, PRJEB18113, PRJEB1829, PRJEB28185, PRJEB39282, PRJEB39330, PRJEB39352, PRJEB39924, PRJEB5462, PRJNA176557, PRJNA210519, PRJNA210523, PRJNA256210, PRJNA312138, PRJNA343262, PRJNA379859, PRJNA422979, PRJNA431934, PRJNA432125; and from the China National GenBank Database (CNGBdb) under project accession: CNP0000189. Illumina BovineHD genotype data is available to download from Edinburgh DataShare, https://doi.org/10.7488/ds/2985.

**Funding:** This research was conducted as part of the CGIAR Research Program on Livestock. ILRI is supported by contributors to the CGIAR Trust Fund. CGIAR is a global research partnership for a food-secure future. Its science is carried out by 15 Research Centers in close collaboration with hundreds of partners across the globe (www.cgiar.org). Some of the work described in this paper was supported by grant BB/H009515/1 awarded jointly by the then UK Department for International Development and the UK Biotechnology and Biological Sciences Research Council (BBSRC) under the Combating Infectious Diseases of Livestock for International Development (CIDLID) program to WIM and PGT. This research was funded in part by the Bill & Melinda Gates Foundation and with UK aid from the UK Foreign, Commonwealth and Development Office (Grant Agreement OPP1127286) under the auspices of the Centre for Tropical Livestock Genetics and Health (CTLGH), established jointly by the University of Edinburgh, SRUC (Scotland's Rural College), and the International Livestock Research Institute (ILRI). This work was also supported by funding from the BBSRC (BBS/E/D/30002275) to JGDP. The funders had no role in study design, data collection and analysis, decision to publish, or preparation of the manuscript.

**Competing interests:** We have read the journal's policy and the authors of this manuscript have the following competing interests: The Centre for Tropical Livestock Genetics and Health (CTLGH) has filed a patent application regarding the use of this locus in improving cattle through methods such as marker assisted breeding.

Through a genetic study of an extended pedigree of Boran cattle that are more than three times less likely to succumb to the disease than matched controls, we identify a region on chromosome 15 of the cattle genome associated with a high level of tolerance to the disease. We show that a nonsense variant in a predicted paralogue of FAS-associated factor 1 (*FAF1*) in this region is also associated with survival in an independent cohort, and is linked to rates of cell expansion during infection. This genetic variant can therefore support marker-assisted selection, allowing farmers to breed tolerant cattle and offers a route to introduce this beneficial DNA to non-native breeds, enabling reduced disease incidence and increased productivity, which would be of benefit to millions of rural smallholder farmers across Africa.

## Introduction

East Coast fever (ECF), a tick-borne lymphoproliferative disease caused by the protozoan parasite *Theileria parva*, is among the biggest natural causes of death in cattle across eastern, central and southern Africa, and is estimated to be responsible for at least 1.1 million deaths per year [1]. Direct economic losses attributed to deaths from ECF have been estimated at ~ US$600 million annually [2,3], and disproportionately affect small-scale low-income households [4]. The potential of *T. parva* to extend its range beyond current areas is likely to increase in the next 100 years due to climate change, as conditions become more favourable for its vector, the brown ear tick *Rhipicephalus appendiculatus* [5]. Cattle breeds not native to East Africa are highly susceptible to ECF, and mortality rates can exceed 95% [6]. The disease is consequently a major barrier to the wider introduction of productive European cattle breeds into large parts of Africa. However, as ECF has exerted an unusually strong selective pressure on East African cattle since their introduction to the region approximately 5000 years ago [7], various local cattle breeds have developed high levels of heritable tolerance to the disease [8,9].

As early as the 1950s, the innate genetic tolerance of *Bos indicus* zebu cattle from ECF-endemic areas of Kenya had been experimentally characterised. In 1953, Barnett [9] transported zebu from ECF-endemic and non-endemic areas of Kenya to a farm free of ECF. Calves subsequently born to these animals were then each exposed to a single *T. parva* infected tick. The offspring of zebu from endemic and non-endemic areas of Kenya showed substantial differences in their ability to survive this infection. Colostral transfer was not the cause, as several studies have shown that antibodies are not responsible for naturally acquired immunity to *T. parva* [10]. More recently in 2005, offspring of East African Shorthorn Zebu (EASZ) originating from endemic areas of Kenya, that were born in a non-endemic area and therefore previously uninfected, showed a significantly higher tolerance to challenge with a *T. parva* stabilate than not only Friesian and Boran cattle, but also EASZ originating from non-endemic areas [8]. Whereas all of the EASZ from the ECF-endemic area of Kenya survived infection, only 60–70% of animals from nearby ECF-non-endemic areas survived the same challenge. These studies highlight that among cattle breeds there is a marked difference in tolerance to infection with *T. parva*, suggesting the frequency of natural tolerance alleles differs between cattle derived from ECF-endemic and ECF-non-endemic areas.

Despite this clear evidence for genetic tolerance among certain local breeds, there has been little work to map the genetic loci driving such tolerance. In this study, we analysed an extended *Bos indicus* (Boran) pedigree, the members of which display markedly higher tolerance to *T. parva* infection than other cattle of the same breed. Through a linkage study, we identify a genomic region significantly associated with this increased tolerance.

## Results

### Tolerance to infection by *Theileria parva* in a *Bos indicus* pedigree

Tolerance within a *Bos indicus* (Boran) pedigree was first observed following a vaccine trial undertaken in 2013 [11], in which four out of six animals that did not succumb to infection by *T. parva* were observed to be first generation descendants of the same Boran bull (3167). This included all of the three unvaccinated survivors in this study. This trial was followed by two further natural field challenge trials in 2014 and 2015 using unvaccinated 1st generation progeny of the same sire alongside unrelated animals, and two subsequent natural field challenge trials in 2017 and 2018 examining 2nd generation offspring (the progeny of male first generation descendants of sire 3167) [12]. The survival data for each trial are shown in Table 1. In summary, 67.9% of the first generation offspring of 3167 and 51.1% of the second generation animals survived the field challenge, compared to 8.7% of the unrelated cattle of the same breed across all field trials. Consistent with the animals showing signs of infection and clinical symptoms (see Methods for further details), body temperatures typically peaked around day 15 following exposure (Fig 1). Comparing only animals that died or survived without intervention, across the five field trials the offspring of sire 3167 were observed to be 3.3 times more likely to survive the exposure to *T. parva* than unrelated controls obtained from the same farm (Mantel-Haenszel relative risk = 3.3, 95% confidence interval = 1.7, 6.44). We calculated the heritability ($h^2$) of the tolerance phenotype among these animals by fitting survival status to sex and trial as fixed effects with sire as a random effect. This model returned an estimated $h^2$ of 0.65 (s.e. = 0.57).

### Survival outcome is significantly associated with a region on chromosome 15

We sought to identify loci underlying the heritable tolerance to *T. parva* within this pedigree. The founder of the pedigree, sire 3167, was no longer available, and so his genotype probabilities (GP) were calculated from the genotype frequencies of his offspring (see Methods). Briefly, we generated whole-genome sequence (WGS) data for 43 animals which included 28 1st generation animals, four animals subsequently identified as likely 3rd degree relatives, and 11 unrelated animals. We genotyped a further 78 animals using the Illumina BovineHD array, bringing the total number of animals to 121 comprising 28 1st generation, 47 2nd generation, and 46 unrelated animals. After merging the datasets, filtering and correcting allele mismatches between the WGS and BovineHD data, simulations were performed to calculate the GP of sire 3167 from the genotypes of all 1st generation offspring. In total 465,938 single-nucleotide variants (SNVs) were analysed and calculated to have a GP > 0.98, of which 97%

**Table 1. Summary of survival outcomes for control and pedigree animals over the course of five field trials.**

| Trial[*] | Controls | | | | Sire 3167 offspring | | | |
|---|---|---|---|---|---|---|---|---|
| | Died | Euthanised | Treated | Survived | Died | Euthanised | Treated | Survived |
| 2013 | 8 | 0 | 1 | 0 | 0 | 0 | 0 | 3 |
| 2014 | 9 | 4 | 0 | 1 | 3 | 0 | 3 | 4 |
| 2015 | 8 | 0 | 2 | 0 | 2 | 0 | 1 | 12 |
| 2017[†] | 2 | 1 | 3 | 1 | 4 | 0 | 3 | 17 |
| 2018[†] | 4 | 0 | 0 | 2 | 12 | 2 | 2 | 7 |
| Total | 31 (67.4%) | 5 (10.9%) | 6 (13%) | 4 (8.7%) | 21 (28%) | 2 (2.7%) | 9 (12%) | 43 (57.3%) |

[*]The first three field trials involved 1st generation offspring of sire 3167, whilst the field trials in 2017 and 2018 (indicated by a [†]) involved 2nd generation offspring.

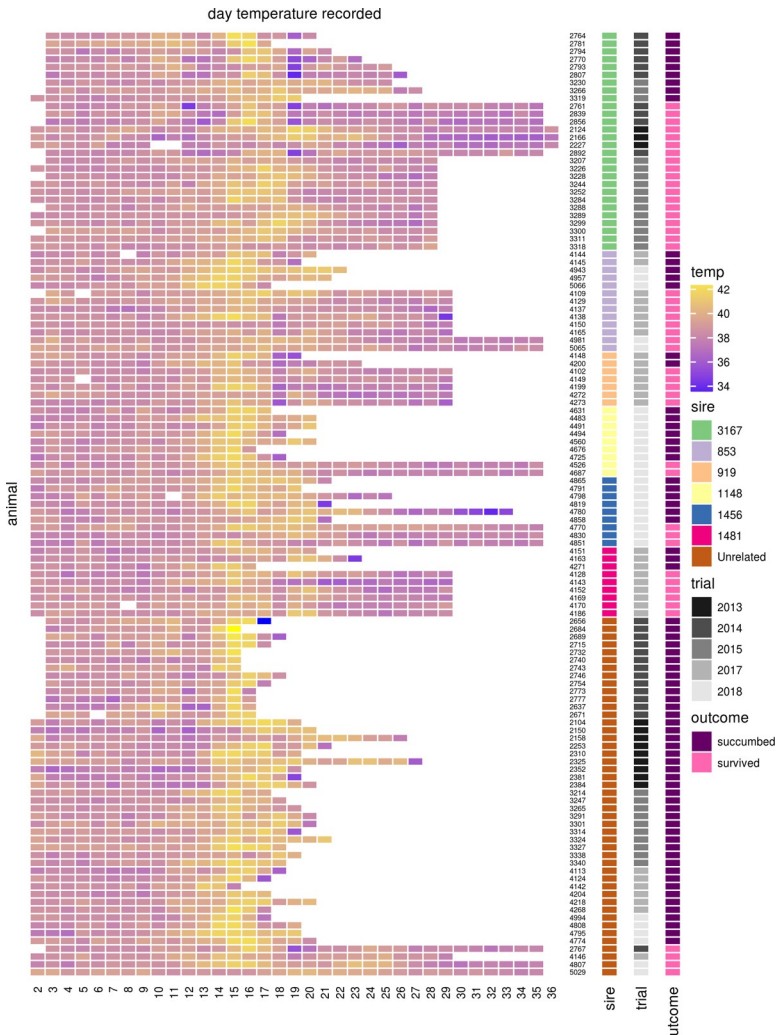

**Fig 1. Heatmap of daily body temperature (˚C) recordings throughout the field trials.** Animals 853, 919, 1148, 1456 and 1481 were all male progeny of sire 3167. Temperature observations commenced on day 2.

(n = 452168) had a GP > 0.9999, i.e. could be inferred with very high confidence. These GP were used to impute genotypes for sire 3167, which were then merged with the genotype data of the pedigree and unrelated animals and collectively phased while accounting for the known pedigree information.

The assumption in this study is that sire 3167 is carrying at least one genetic locus that confers tolerance to *T. parva* infection. The pedigree could therefore inherit a tolerance haplotype from this founding bull, but we did not want to exclude the possibility of it also being inherited down the maternal line. To do this, haplotype analyses were conducted by first partitioning the phased genotype data into 1 Mb windows, for which the Hamming distance of each animal's paternal and maternal haplotypes to each of those of sire 3167 was calculated, resulting in four distance values per window for each animal. We fitted a regression model with binomial survival outcome as a response to the four haplotype distances alongside sex, trial ID and sire to account for these potential confounders. This allowed us to determine whether animals carrying particular haplotypes that derived from, or that were highly similar to those of, sire 3167 were more likely to survive exposure to *T. parva* infection.

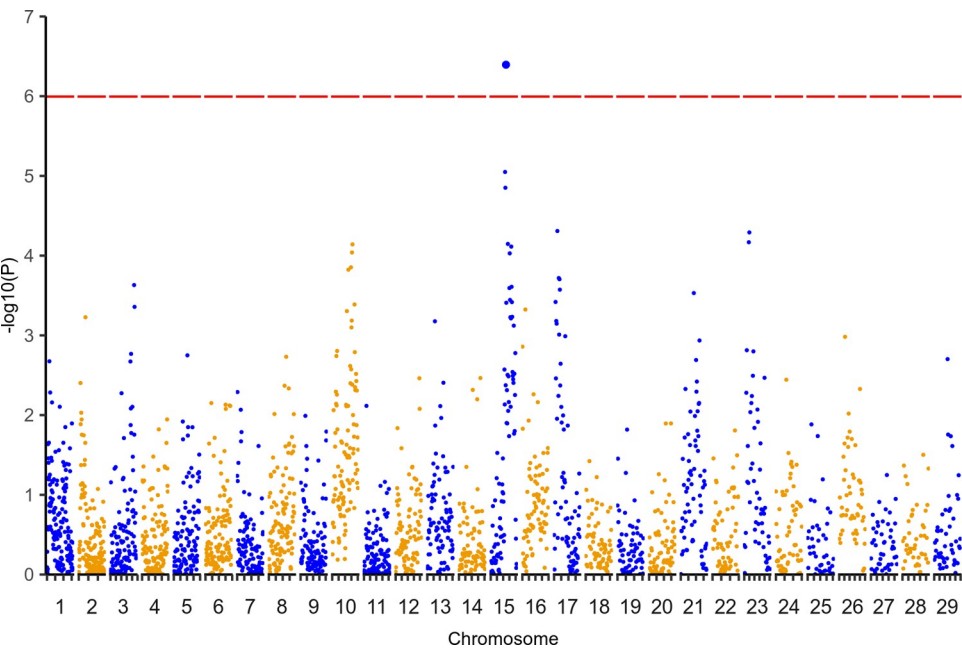

**Fig 2. Regression analyses on haplotype distances.** Results of regression analyses of haplotypes versus survival status. A single window crossed the corrected significance threshold determined from 1000 permutations, indicated by the dashed red line at p < 0.05.

We identified a genome-wide significant association (p = 4.12 x $10^{-07}$, adjusted for multiple testing via permutations p = 0.027, see Methods) on chromosome 15 (15:49–50 Mb; Fig 2), followed by two neighbouring haplotypes (15:46–47 Mb unadjusted p = 9.1 x $10^{-6}$; 15:47–48 Mb unadjusted p = 1.43 x $10^{-5}$) which, although also nominally significant, were not significant genome-wide after correction (15:46–47 Mb corrected p = 0.23; 15:47–48 Mb corrected p = 0.32). The paternally-derived 1 Mb haplotype windows of surviving animals in this region (15:46–50 Mb) were more similar to both the A (median Hamming distance $DA = 0.069 ± 0.165$) and B ($DB = 0 ± 0.160$) haplotypes of sire 3167 than those of the animals that succumbed to infection ($DA = 0.254 ± 0.250$; $DB = 0.278 ± 0.258$). The region was screened for evidence of structural variations (SVs) in animals for which WGS data was available, using Delly (https://github.com/dellytools/delly), however, none of the putative SVs identified were significantly correlated with survival outcome.

## Characterisation of variants within the tolerance locus

Having identified a genomic region significantly associated with survival outcome we next sought to identify potential candidate functional variants within the locus for further investigation. To do this we characterised both the impact of variants on coding regions as well as any potential link to the expression of nearby genes. To define potential regulatory variants (eVariants) we sequenced mRNA extracted from white blood cells collected from 29 animals (n = 23 pedigree, n = 6 unrelated) involved in the 2018 field trial at day 0 (i.e. before their transport to Ol Pejeta Conservancy), day 7 and day 15. Within the target region, 720 variants showed a nominally significant association between their genotype and the expression level of a nearby gene on at least one of the timepoints (S1 Table), and a further 587 variants were tagged as potential response eQTLs, where the association between genotype and gene expression

differed between timepoints (S5 Table). However, none of these associations were significant after applying a false discovery rate correction to account for multiple testing.

We next characterised the impact of the variants in this region on predicted coding regions. The allele frequency and functional consequence of all SNVs spanning 1 Mb upstream and downstream of the identified wider region (15:45–51 Mb) were characterised from the whole-genome sequence (WGS) data. The functional consequences of variants were determined with the Ensembl Variant Effect Predictor (VEP) that annotates variants according to their predicted impact on coding regions [13]. After filtering to retain high-quality variants, we identified 58066 variants and 235 genes in this region. These included 57119 single nucleotide variants, 500 indels (an insertion or deletion, affecting two or more nucleotides), and 447 sequence alterations (sequence ontology SO:0001059). A summary of all predicted variant consequences is provided in the supporting information (S6 Table). From these, we selected 906 variants predicted to have a functional consequence with a moderate or high impact. As any variant conferring tolerance to infection is unlikely to be present in non-native breeds, as evidenced by their extremely high mortality when exposed to *T. parva*, we removed variants with an alternate allele frequency (aAF) > 0.1 across European taurine breeds, leaving 531 variants. Similarly, we expected to observe the variant at a high frequency in the 1st generation offspring of sire 3167 that survived infection, and so removed variants with aAF < 0.5 in these animals, resulting in 52 variants of which 4 were predicted to be high impact (Fig 3A): 1 frameshift variant, 2 stop gained and 1 start lost, each affecting a different gene. Each of these variants had an alternate allele frequency < 0.04 in European taurine, < 0.091 in African taurine breeds, but ranged from 0.22 to 0.54 in African and Asian indicine breeds, and from 0.53 to 0.78 in the pedigree of 3167 (Fig 3B).

Of the genes associated with these high impact variants, two are olfactory receptors (*OR51H1* and *OR51G1*) and two are novel genes—the first is a paralogue of haemoglobin subunit beta (*HBB*) and the second a paralogue of FAS-associated factor 1 (*FAF1*). It should be noted we could not find any evidence of any of these genes being expressed in white blood cell RNA-seq data, suggesting this is not their tissue of action. Olfactory receptors (ORs) have been found to be involved in immune responses associated with the olfactory bulb, and there is increasing evidence of OR expression in non-olfactory tissues, however, there remains a significant gap in our knowledge concerning the functional importance of ectopic ORs [14]. *HBB* is associated with erythrocytes and variants in *HBB* have previously been linked to protection to human malaria, caused by another pathogenic Apicomplexan protozoon, *Plasmodium falciparum*. However, as infection of lymphocytes is responsible for the pathology of disease caused by *T. parva* and infected lymphocytes are also the target of the protective acquired immune response, with infection of erythrocytes having little or no role in either of these, the role of *HBB* tolerance to *T. parva* infection is not immediately obvious. In contrast, *FAS* is a member of the TNF-receptor superfamily involved in apoptosis, and *T. parva*-infected cells have previously been shown to be resistant to *FAS*-induced cell death [15]. Thus, given the known association of *FAS* with Theileria, we focused on further examining the association of the high impact variant in the *FAF1* paralog with tolerance to *T. parva* infection.

## Genotyping of the *FAF1* paralog confirms a significant association with survival outcome

The *FAF1* paralog shares 96.75% identity with 68.77% of *FAF1*'s nucleotide sequence. The stop gain variant (NC_037342.1:g.50823230C>T) in the *FAF1* paralog (referred to here as *FAF1B*) converts an arginine codon (cga) to a stop codon (tga) which would result in a truncated protein. Within *FAF1* itself there is no variant recorded in dbSNP at the homologous base position

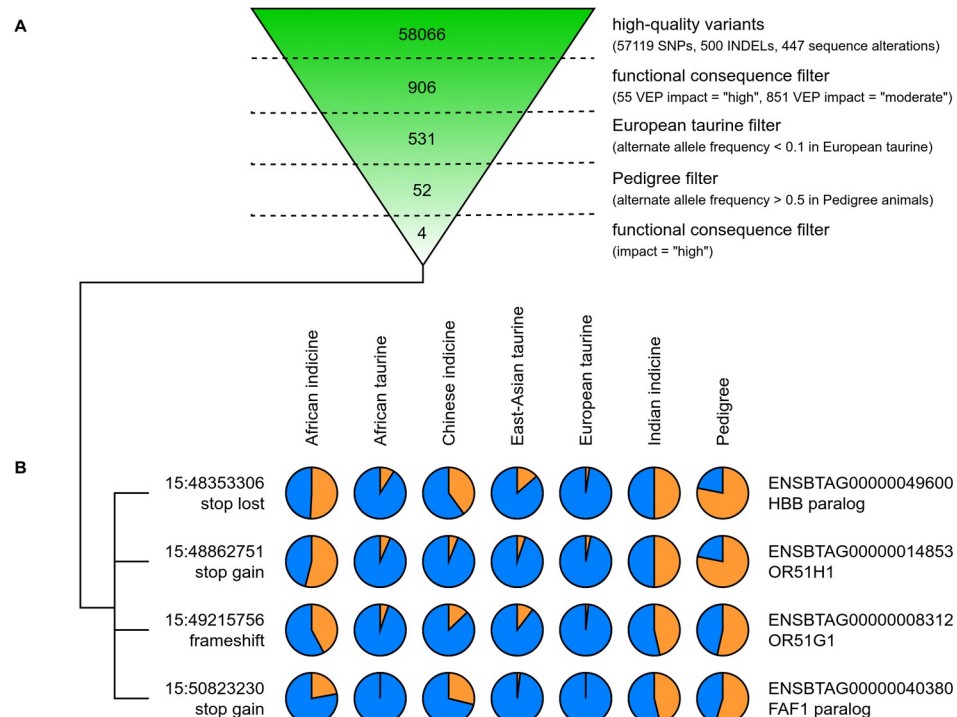

**Fig 3. Prioritisation of variants.** (A) Successive filters were applied to the high-quality variants in the region identified as significantly associated with tolerance to *T. parva* infection (15:45–51 Mb). This reduced the number of variants from 93960 to four predicted to have a high functional consequence by the Ensembl Variant Effect Predictor (VEP), a low alternate allele frequency (aAF < 0.1) in European taurine animals, and a high alternative allele frequency in Pedigree animals (aAF > 0.5). (B) The allele frequencies (reference allele in blue, alternative allele in orange) of these four variants were subsequently calculated in different populations (145 European taurine animals, 44 East-Asian taurine, 35 African taurine, 112 African indicine, 64 Chinese indicine, and 36 Indian indicine), indicating the alternative allele is potentially of an indicine origin.

(3:95786449) to the *FAF1B* variant (15:50823230). The *FAF1B* variant is not present on the BovineHD array used to genotype the majority of the animals. To establish if there is a significant association between genotype and survival outcome across the wider set of animals, we designed primers to amplify and sequence this variant in 57 trial animals that were not previously whole-genome sequenced, bringing our genotype data for *FAF1B* to 100 animals. These animals include 77 pedigree (43 survived, 34 succumbed to infection) and 23 unrelated individuals (4 survived, 19 succumbed to infection). In all, 88% of the animals homozygous for the variant allele (T/T) survived the field challenge (7 out of 8, the final animal having been treated rather than having died from the infection), compared to 62% (36 out of 58) of heterozygotes (C/T) and only 12% (4 out of 34) of those homozygous for the reference allele (C/C; Table 2). Analysis by logistic regression of survival status versus genotype at this variant was significant across all animals (likelihood ratio test p = 2.48 x $10^{-04}$), within unrelated animals (likelihood ratio test p = 5.38 x $10^{-04}$), and within the pedigree (likelihood ratio test p = 6.34 x $10^{-04}$), when controlling for sex, trial, sire (where pedigree animals were involved), including 3rd degree relatives in the pedigree, and classifying animals that received treatment or were euthanised as having died.

Considering explicitly the unrelated and pedigree (1st and 2nd generation) animals that survived or succumbed to infection, a comparison of survival curves fitting survival time versus genotype at the *FAF1B* variant while accounting for degree of relatedness returns a significant

**Table 2. The genotype counts of the *FAF1B* variant.**

| Genotype | Status | Unrelated | 3rd degree* | 2nd gen. | 1st gen. | Total |
|----------|--------|-----------|-------------|----------|----------|-------|
| C/C | Succumbed | 13 | 2 | 7 | 0 | 22 (64.7%) |
| C/C | Euthanised | 2 | 0 | 1 | 0 | 3 (8.8%) |
| C/C | Treated | 3 | 0 | 2 | 0 | 5 (14.7%) |
| C/C | Survived | 0 | 0 | 4 | 0 | **4 (11.8%)** |
| C/T | Succumbed | 1 | 0 | 9 | 5 | 15 (25.9%) |
| C/T | Euthanised | 0 | 0 | 1 | 0 | 1 (1.7%) |
| C/T | Treated | 0 | 0 | 3 | 3 | 6 (10.3%) |
| C/T | Survived | 3 | 2 | 16 | 15 | **36 (62.1%)** |
| T/T | Succumbed | 0 | 0 | 0 | 0 | 0 (0%) |
| T/T | Euthanised | 0 | 0 | 0 | 0 | 0 (0%) |
| T/T | Treated | 0 | 0 | 0 | 1 | 1 (12.5%) |
| T/T | Survived | 1 | 0 | 2 | 4 | **7 (87.5%)** |

*Upon genotyping, four controls were determined most likely to be 3rd degree relatives of sire 3167. These are separated out in the above table. Note there were no homozygote C/C first generation animals as sire 3167 was T/T at this variant.

association (log-rank p = 4.13 x $10^{-13}$; Fig 4A). Further to its association with survival, among these animals that succumbed to the disease, those heterozygous at the variant succumbed significantly (Mann-Whitney U test p = 7.3 x $10^{-04}$) later on average than homozygous reference animals (mean of 21.27 ± 2.25 days versus 18.55 ± 1.7 days; Fig 4B). An animal's genotype at this variant was also observed to be associated with their temperature over the course of the infection (Fig 4C), with T/T animals generally showing a lower increase in temperature, and in particular at days 15 and 16 (ANOVA F test P of 2.44 x $10^{-06}$ and 4.43 x $10^{-07}$, respectively, when accounting for relatedness, sex and trial; Fig 4D).

To determine how much variance in heritability is explained by the *FAF1B* SNV we ran linear mixed effect models using residual maximum likelihood to estimate parameters. Having calculated the heritability ($h^2$) based on phenotypic data alone as 0.65 (s.e. = 0.57) as stated above, we subsequently accounted for genetic effects by fitting the genotype at the *FAF1B* SNV as a fixed effect. This model returned a significant additive SNV effect of $\alpha$ = 0.43 ± 0.083 for a single copy of allele T. This is highly significant assuming an additive model $t$ = 5.2 (Student's left-tailed t-distribution p = 2.51 x $10^{-06}$), with no evidence for a dominant effect (dom = 0.12 ± 0.09) $t$ = 1.25 (Student's left-tailed t-distribution p = 0.18). The SNV is estimated to explain 31.9% of the phenotypic variance among these animals.

## *FAF1B* genotype-associated tolerance replicated in independent population

We next investigated any evidence for replication of this association in the independent Infectious Disease of East African Livestock (IDEAL) cohort. The IDEAL study was conducted in a *T. parva*-endemic area in western Kenya in 2007 in a smallholder system free of buffalo, and involved the intensive monitoring of 548 East African Shorthorn Zebu (EASZ) calves for the first year of life. The study generated data on the frequency and clinical signs of infections and their impact on health and growth [16], with 32 calves determined, by clinical signs and post mortem observations, to have died of ECF. There were a number of important differences between this study and the field trials reported above. The animals in the IDEAL study were spread across farms and different environmental exposures and, unlike the Boran breed, EASZ are believed to exhibit generally elevated tolerance to ECF—supported by the IDEAL project reporting a 6% mortality due to *T. parva* infection despite 76% of animals becoming infected

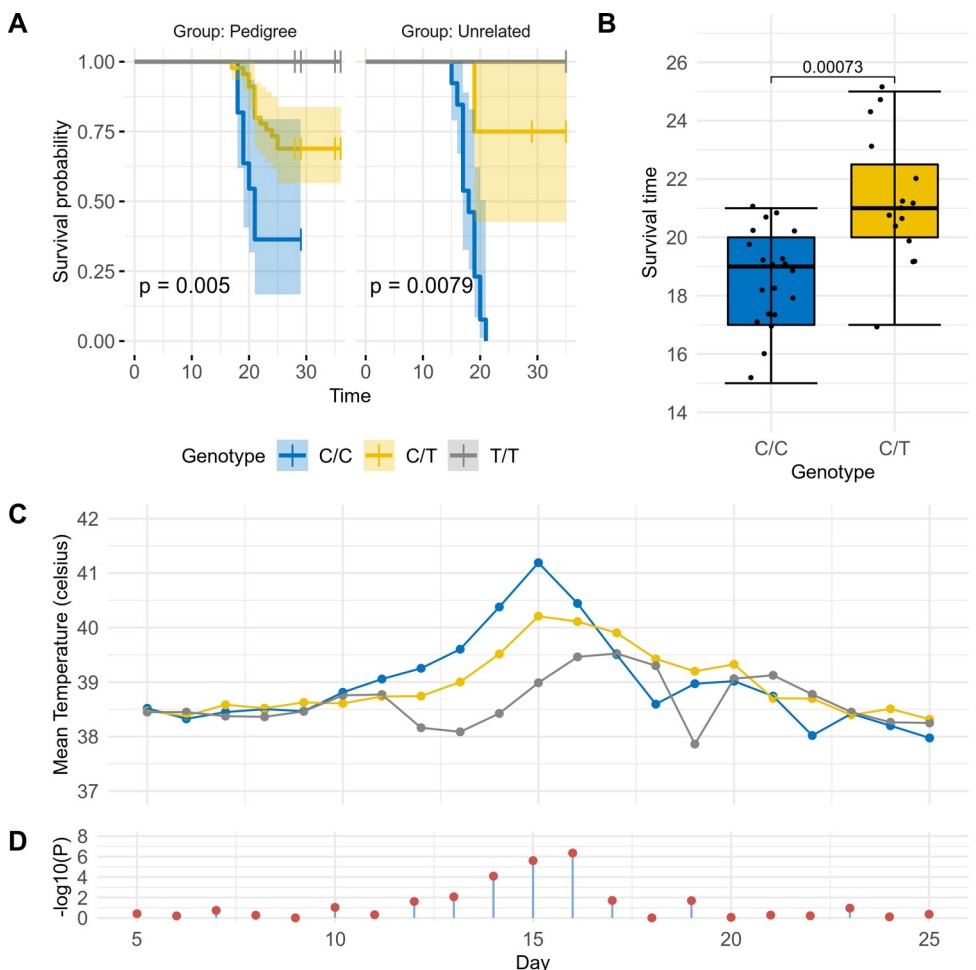

**Fig 4. Survival analyses for the *FAF1B* variant in pedigree (1st and 2nd generation) and unrelated animals.** (A) Analysis of survival curves in animals that succumbed or survived *T. parva* infection reveals significant associations between survival probability and genotype. Log-rank p values are reported. Comparing survival curves when fitting survival time against genotype whilst accounting for relatedness (unrelated, 1st generation, 2nd generation) returns log-rank p = 4.13 x 10$^{-13}$. (B) Comparison of survival time versus genotype for animals that succumbed to infection. The mean survival time for C/C animals was 18.55 ± 1.7 days, while mean survival time for C/T animals was 21.27 ± 2.25 days. A Mann-Whitney U test p value is reported. (C) Mean temperature versus field trial day for all animals regardless of survival outcome and relatedness, grouped by genotype. Temperature peaked at day 15 in C/C and C/T animals, and at day 17 in T/T animals. (D) Fitting T allele count against temperature whilst accounting for relatedness, sex and trial, returns significant associations from days 13 to 16 (ANOVA F-test p < 0.01).

in their first year of life [17]. Furthermore, transmission of *T. parva* in the IDEAL study will have been via tick transmission from cattle to cattle, rather than transmission to cattle from ticks that had previously fed on buffalo as in the field trials. We genotyped 130 EASZ at the *FAF1B* SNV using the same primers as for the Boran. This included the 32 calves that died of ECF, and 98 randomly selected survivors from the IDEAL study. Consistent with the field trial results suggesting the T allele is associated with tolerance, no homozygote T/T animals in the study succumbed to ECF (Table 3). Logistic regression of survival status versus genotype while accounting for the animal's sex and the latitude, longitude, and elevation of the sampling location was significant in this cohort of 130 animals (likelihood ratio test p = 0.029) with the same direction of effect as in the 3167 pedigree.

**Table 3. IDEAL cohort genotypes.**

| Genotype | Succumbed to ECF (n = 32) | Survived† | |
|:---:|:---:|:---:|:---:|
| | | **Episode (n = 31)** | **No-episode (n = 67)** |
| C/C | 16 (50%) | 21 (68%) | 30 (45%) |
| C/T | 16 (50%) | 8 (26%) | 27 (40%) |
| T/T | 0 (0%) | 2 (6.5%) | 10 (14.9%) |

†Randomly selected animals that survived are further categorised according to whether or not they presented any evidence of a clinical episode associated with ECF, which include: pyrexia, and hypertrophic parotid and prescapular lymph nodes.

### *FAF1B* genotype is associated with the rate of *in vitro* proliferation of infected cells

After infection *in vitro*, cells from surviving cattle proliferate more slowly than those from susceptible cattle [12]. We therefore sought to determine if there is also a difference in the *in* vitro expansion of *T. parva*-infected cells based on the *FAF1B* genotype of the animals. Peripheral blood mononuclear cells (PBMCs) were cultured and infected by incubation with salivary glands dissected from *R. appendiculatus* fed on animals infected with *T. parva*. The level of infection in ticks was estimated by counting the number of infected acini in a sample of dissected salivary glands, and the sporozoite suspension adjusted to a concentration equivalent to 2000 infected acini per ml (see [18]). Equal volumes of sporozoites and cells ($2 \times 10^7$ PBMCs) were cultured from 12 cattle unrelated to 3167 carrying the C/C genotype at the *FAF1B* SNV, and 12 cattle carrying the T/T genotype, which comprised six animals from the tolerant pedigree and a further six unrelated animals for comparison. Counting of cells stained with trypan blue every 2 days over a period of 12 days revealed a significant association between live cell count and genotype when fitting day as an interaction term and accounting for relatedness (see Methods, F-test p = $1.42 \times 10^{-07}$; Fig 5A). To determine if there was a difference in infectivity associated with the *FAF1B* SNV, cells were stained with a fluorophore directed at the polymorphic immunodominant molecule (PIM) expressed on the *T. parva* schizont's surface. As with the live cell count, a significant association between the proportion of infected (PIM+) cells and genotype was observed when fitting day as an interaction term and accounting for relatedness (F-test p = $7.56 \times 10^{-09}$; Fig 5B).

Consequently, these data suggest the association of the *FAF1B* locus with survival in the 3167 and IDEAL cohorts may also be reflected in different capacities of *T. parva*-infected cells to establish and expand *in vitro*.

## Discussion

Few studies have reported naturally occurring loci of large effects that confer resistance to disease in livestock. Of those for which potential causal variants have been identified, the infectious agent is generally viral in nature. For example in chicken a range of variants have been associated with variable resistance to myxovirus [19], and avian sarcoma and leukosis viruses subgroups A [20–22], B [23] and C [24], affecting *MX1*, *CD320*, *TNFRSF10B*, and *BTNA1*, respectively. In sheep, resistance to lentivirus has been associated with variants affecting *TMEM154* and *ZNF389* [25,26]. In the case of the latter, White *et al.* [26] identified a 2 bp deletion in the 5' genomic region of ZNF389 that returned a significant association with proviral concentration across multiple breeds. Here, we report in cattle, the significant association of a genetic variant with tolerance to infection by an Apicomplexan parasite, *T. parva*.

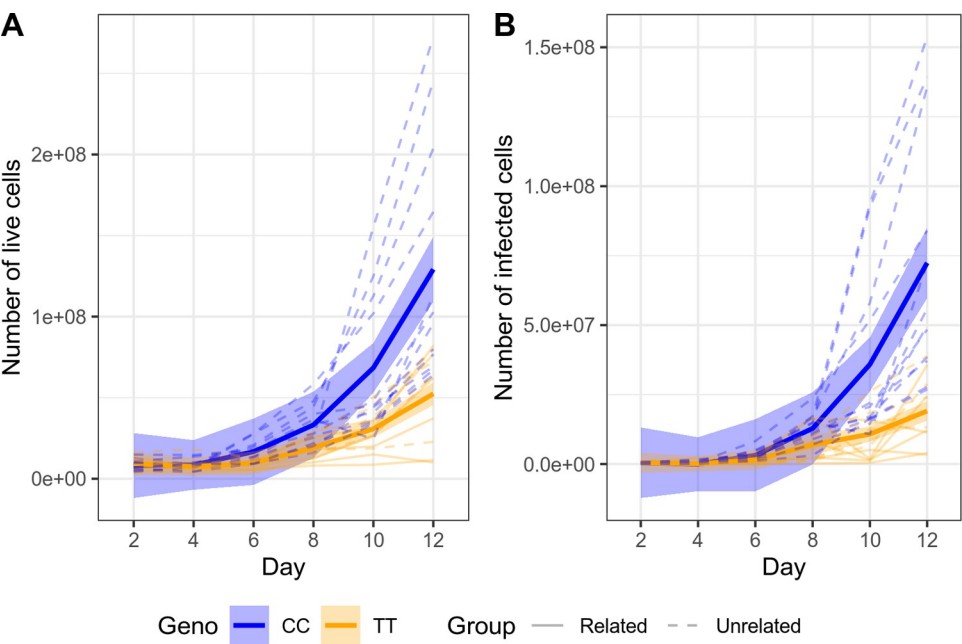

**Fig 5. Cell expansion and infectivity levels in cultured cells infected with *T. parva*.** (A) Animals possessing a C/C genotype at the *FAF1B* SNV exhibit higher live counts throughout the course of infection compared to T/T animals. The association between live cell count and genotype when fitting day as an interaction term and accounting for relatedness is significant (F-test p = 1.42 x $10^{-07}$). (B) In addition, animals possessing a C/C genotype exhibit a higher proportion of infected (PIM+) live cells, which is also significant (F-test p = 7.56 x $10^{-09}$). Thin dashed lines indicated unrelated animals, solid, thin lines indicate animals from the ECF-tolerant pedigree. The blue and orange ribbons and thick blue and orange lines indicate the 95% confidence interval around the mean for C/C and T/T genotypes, respectively. The data underlying these figures is provided in the supporting information (S7 Table).

The progeny of an extended pedigree, sired by founding bull 3167, demonstrate a clear, heritable tolerance to buffalo-derived *T. parva* infection, which haplotype-based analyses reveal to be significantly associated with an extended 6 Mb genomic region on chromosome 15. Notably, this same region was one of a small number showing evidence of between-breed selective sweeps in both cattle and water buffalo in a recent study [27], and both species are susceptible to a closely related *Theileria* species, *T. annulata*. This, along with the elevated non-reference allele frequency among indicine cattle breeds at variants in this region, raises the possibility that the tolerance phenotype first arose as a response to *T. annulata*, which is transmitted by a different tick and is present in a region extending from north Africa and southern Europe into the Middle East and Asia. This was then introduced into Africa in one of the waves of introduction of *Bos indicus* breeds. Within this genomic region we identified a putative stop-gained variant within a paralogue of the *FAF1* gene, *FAF1B*, the genotype of which is significantly associated with survival outcome. As Fas-induced apoptosis has previously been shown to be targeted by the parasite to facilitate the transformation of T cells [15], this gene is a plausible candidate for a central role in mediating tolerance. We also find the *FAF1B* SNV to be associated with cell expansion and infection levels *in vitro*. A key hallmark of *T. parva* infection is the rapid proliferation of infected host lymphoctyes, and, although further work is required, these *in vitro* findings potentially support the idea that the tolerance phenotype is linked to limiting the proliferation of infected cells [12].

It should be noted we could find no evidence that *FAF1B* is expressed in white blood cells (WBCs) before or after infection, though it may be active in other tissues. Although we found little evidence for any eQTLs in this region potentially driving the phenotype, this analysis had

limitations in that the sample size was small, blood is potentially confounded by cell-type composition differences, and, although a reasonable candidate, the WBC population may not be relevant. Consequently, further work is required to elucidate any causal relationship between this *FAF1* paralogue and the enhanced tolerance to *T. parva* linked to this locus. Possible experiments could include testing the susceptibility of *FAF1B* C/C versus T/T cells before and after infection to Fas-mediated apoptosis, and using CRISPR/Cas9 to introduce candidate tolerance alleles into cells isolated from susceptible cattle and subsequently test their response to infection. Irrespective of whether it is driving tolerance though, we have demonstrated the putative stop gained variant in this gene is strongly linked to the tolerance phenotype and importantly provides a potential effective genetic marker for breeding for tolerance. Notably, of the combined total of 20 animals that were homozygous for the alternate allele at this locus across the Boran pedigree, controls and EASZ, none succumbed to *T. parva* infection (though one was treated). This is in contrast to the 53% of animals which were homozygote for the reference allele (n = 44) that succumbed (n = 83). Although a number of these animals were related, the lack of deaths among homozygote carriers adds support to the promise of potential breeding for tolerance using this allele. It is also worth noting that of the four identified 3rd degree relatives, only the two that possessed a T allele survived infection. Furthermore, those heterozygote carriers that succumbed to the disease were observed to survive several days longer on average than animals carrying no copies of the variant, and to display some evidence of potentially showing fewer clinical episodes in the IDEAL cohort. *T. parva* can be transmitted via ticks that have previously fed on either infected cattle or infected African buffalo. The challenge in the field trials was buffalo-derived, as cattle had been absent from the trial site prior to the introduction of the trial animals. In contrast the challenge in the IDEAL study was cattle-derived. Consequently, the evidence points towards this locus being potentially protective against disease caused by both forms of the parasite.

In summary, we have therefore validated a genetic marker which is significantly associated with survival outcome to *T. parva* infection. This marker can consequently be applied to support selective breeding programmes with a view to improving the tolerance of cattle populations, in order to reduce the substantial impact of *T. parva* on sub-Sahara African countries. Confirmation of the functional variant in this region will facilitate the application of modern gene editing technologies to formally demonstrate the functional role of the variant and ultimately to rapidly increase its frequency in imported, productive cattle throughout Africa.

## Methods

### Ethics statement

The study protocols were approved by ILRI's Institutional Animal Care and Use Committee (References 2013–03, 2014–32, 2015–29, 2017–02, 2018–10).

### Field trials

Field trials involved transporting animals from ILRI's Kapiti Research Station in Machakos county, a region largely free of ECF, to the Ol Pejeta Conservancy (Kenya) in a region endemic for *T. parva*. The study site where animals remained for up to 35 days was free of other cattle but populated by African buffalo [11]. The 2013, 2014 and 2015 field trials involved 1st generation progeny of sire 3167 while the 2017 and 2018 field trials involved progeny of males sired by 3167; in all trials unrelated animals were also included, and researchers conducting the phenotyping were unaware as to which animals were pedigree or unrelated. Although pedigree records were maintained throughout, for the 2017 and 2018 field trials animals were genotyped in advance to verify their relatedness. Genetic analyses (described below) identified 21

individuals that were incorrectly documented as belonging to the pedigree of sire 3167; eight pedigree individuals whose sire was incorrectly recorded, but which remained within the pedigree; and seven individuals thought to be unrelated but which were found to be 3$^{rd}$ degree relatives from four sibships. Animals in all field trials were screened in advance to ensure they tested negative by ELISA [28] for antibodies against *T. parva* and, for the last four trials, *T. mutans*, and in addition in the 2013 [11], 2017 and 2018 field trials by p104 PCR [29] to test for active *T. parva* infection. During the field trials body temperature recordings were taken on a daily basis. On a small number of occasions a recording could not be taken due to the temperament of the animal. The first four days of temperature recordings were excluded from downstream analyses as these typically exhibited unusual deviations due to the stress of transport. A summary of phenotypic data for field trial animals analysed in this study is provided in S1 Table, along with temperature observations in S2 Table.

The most likely cause of death for the first study was reported to be buffalo-derived *T. parva* infection, based on clinical signs and post mortem observations [11]. Similar observations were made in the subsequent studies. All animals exhibited pyrexia except one (4150; Fig 1), and swollen parotid and prescapular lymph nodes, whilst laboured breathing, nasal discharge and corneal opacity were commonly observed. Microscopic examination of aspirates from the parotid or prescapular lymph nodes revealed *T. parva* macroschizonts in at least one lymph node of every animal. Routine post mortem observations included frothy exudate in the trachea and pulmonary oedema. Seroconversion to anti-*T. parva* antibodies was detected in 46 of the 47 surviving animals and in 42 of the 74 animals that succumbed.

Survival risk ratios were calculated from a fixed-effects (Mantel-Haenszel) meta analysis of animals that survived or died, without intervention, using the meta.MH function of the rmeta package for R. To address zero values for unrelated survivors in field trials 2013 and 2015, a zero-cell correction was applied by adding 0.5 to all values [30].

## Sampling

Blood was collected from the animals in EDTA tubes and extracted using DNEasy Blood and Tissue Kits (Qiagen) according to manufacturer's instructions. For genotyping on the Illumina BovineHD BeadChip, DNA integrity was first assessed by electrophoresis on a 1.5% agarose gel, and quantitated by NanoDrop spectrophotometer for dilution to a concentration of 50 ng/μl in 40μl of elution buffer. DNA samples were shipped to Edinburgh Genomics where quality was assessed by PicoGreen assay prior to genotyping. For whole-genome sequencing DNA was extracted using the MagNA Pure LC Total Nucleic Acid Isolation Kit (Roche Diagnostics GmbH).

## Survival analysis

Survival analyses were performed and plotted in R [31] (v3.4.4) using the survminer (v0.4.6) package. Briefly, a survival object was constructed excluding animals that were treated or euthanised fitting survival time, defined in days as either the end of the field trial or the day on which the animal died, against genotype whilst accounting for degree of relatedness (unrelated, 1$^{st}$ or 2$^{nd}$ generation). P values were calculated from the fitted survival curves using the log-rank method.

## Processing of whole-genome sequence data

Sequencing was performed on the Illumina HiSeq X platform using 150bp pair end reads with 550bp insert, TruSeq PCR free libraries, to a target depth of 30X. Sequence reads were aligned to the ARS-UCD1.2 assembly using BWA-MEM [32] (v0.7.17), and processed using GATK

[33,34] (v4.0.8.0) applying MappingQualityNotZeroReadFilter, MarkDuplicates, and base quality score recalibration (BQSR) using known sites from the 1000bulls genome project (http://www.1000bullgenomes.com/). Variants were called using HaplotypeCaller [35,36] to generate gVCF files, from which joint variant calling was performed across samples, followed by variant quality score recalibration (VQSR).

### Processing of BovineHD genotyping data

Genomic coordinates for the Illumina BovineHD Genotyping BeadChip were updated from the UDM3.1 reference assembly to the newer ARS-UCD1.2 assembly using data provided by Robert Schnabel from the University of Missouri (UMC_marker_names_180910.zip, available at https://www.animalgenome.org/repository/cattle/UMC_bovine_coordinates/). The array data was converted from PED to VCF format using Plink [37] (v1.90blg). A Python script was written to cross-reference the alleles for SNVs from the genotype data against those from the sequence data, to correct for discrepancies in reference allele and/or strand between the different datasets. SNVs with alleles whose concordance could not be validated (A/T, T/A, C/G, G/C) were discarded. The VCFs from genotyping and sequencing were subsequently merged using bcftools [38] (v1.2), and filtered to remove SNVs with a minor allele frequency (MAF) < 0.05 and call rate < 0.95.

### Analysis of relatedness

To cross-validate the written breeding records and ensure that individuals were not assigned to the pedigree of sire 3167 incorrectly, the genotype data were analysed using VCFtools [39] (v0.1.13) to calculate relatedness as per Manichaikul et al. [40]. Relatedness values range from $\Phi < 0.0442$ for unrelated individuals, $0.0442 < \Phi < 0.0884$ for 3rd degree relatives, $0.0884 < \Phi < 0.177$ for 2nd degree relatives, and $\Phi > 0.1777$ for 1st degree relatives.

### Imputation of founding sire 3167

Genotypes from the 1st generation offspring were phased with SHAPEIT2 [41] (v2r837) using a genetic map estimated from physical SNV positions assuming 1 cM/Mb, employing a window size (-W) of 5 Mb, and the duoHMM [42] (v0.1.7) algorithm to take into account the available pedigree information. A Python script was written to estimate sire 3167's genotype probabilities from the 1st generation genotypes. For each SNV, 1 million simulations were performed to generate genotypes for 34 individuals (the number of 1st generation offspring genotyped). During each iteration, for each individual a 'paternal' allele was randomly chosen from each of sire 3167's possible genotypes {0/0, 0/1, 1/1}, and combined with a random 'maternal' allele {0, 1}. Thus, for each iteration we recorded the counts of offspring for each combination of alleles {0/0, 0/1, 1/1} for each of sire 3167's possible genotypes. The simulations incorporated a 5% error rate which would cause the paternal allele to be switched. From these simulations, the offspring genotype counts that matched those of the 1st generation data were retrieved for each of sire 3167's possible genotypes, and the frequency of each recorded as sire 3167's genotype probabilities. Sire 3167's genotypes were then imputed from these probabilities and the 1st generation phased genotype data in 5 Mb windows using IMPUTE2 [43] (v2.3.2). The imputed genotypes for sire 3167 and the 1st generation genotype data were then combined with the 2nd generation genotype data, and collectively phased using SHAPEIT2 and duoHMM.

## Haplotype association analyses

Assuming sire 3167 carries one or more haplotypes associated with tolerance to *T. parva*, we partitioned the genotype data into 1 Mb 'haplotype blocks' and calculated the Hamming [44] distance for each progeny's paternal (A) and maternal (B) haplotypes (h) to those imputed for sire 3167 (r). This resulted in four haplotype distance metrics per individual {$A_h,A_r$; $A_h,B_r$; $B_h$, $A_r$; $B_h,B_r$} which were divided by the number of SNVs in the haplotype to correct for the varying SNV density of haplotypes. The analysis was performed in R using the proxy (v0.4–22) package. Multiple regression was performed in R to test binomial survival status against the four haplotype distances with sex, field trial and sire fitted as covariates. This model was then compared to a reduced model without the four haplotype distances included using ANOVA. Individuals that received veterinary intervention due to their severe symptoms, or were euthanised, were treated as non-survivors for the purpose of this binary phenotype. To determine significance thresholds that account for multiple testing, 1000 permutations were performed where the set of four distances were permuted between individuals. For all blocks on the same chromosome the distances were swapped between the same sets of animals to maintain their relationship across regions. Thus, the phenotypic metadata remained associated with the correct ID and any linkage between haplotypes along a chromosome remained intact. The minimum p value observed across all blocks in the genome for each permutation was recorded, from which a 0.05 significant threshold was determined as the value where just 5% of these values were smaller (S3 Table).

## Expression quantitative trait loci (eQTL) analyses

RNA was extracted from white blood cells of animals sampled during the 2018 trial prior to being transported to the field site (day 0) and on days 7 and 15 of the field trial. These animals included 15 that succumbed to infection, 9 that survived, 2 that were euthanised and 2 that were treated. Following phenol chloroform extraction, mRNA was sequenced on the Illumina HiSeq platform to generate ~70M x 50 bp reads per sample. RNA sequencing reads were aligned to ARS-UCD1.2 using STAR [45] (v2.7.1a;). Stranded fragments per kilobase of exon model per million reads mapped (FPKM) values were calculated for exon features from the alignments using Htseq-count [46] (v0.11.2;). FPKM values were tabulated and filtered in R for each sampling day to retain only genes where at least 50% of samples had FPKM $\geq$ 3. Response expression quantitative trait loci (reQTL) analyses were performed in R as follows. Genes within the region 15:44–52 Mb were identified, and for each day the FPKM of each gene for each individual was retained, along with the allele dosages for *cis* variants—these included any bi-allelic SNV with MAF > 0.1 within 1 Mb upstream of the gene's start position to 1 Mb downstream of its end position. SNV allele dosages were derived from the genotype data generated using the BovineHD array described above. Animals were assigned a group value of 'pedigree' or 'control', and for each day we regressed a gene's FPKM values against the allele dosages of each of its *cis* variants independently, while accounting for an individual's sex and group assignment. To identify reQTL we used a beta-comparison approach. Here we performed pairwise comparisons of regression slopes for an eQTL at the different time points in a z-test:

$$Z = \frac{\beta_x - \beta_y}{\sqrt{\delta_x^2 + \delta_y^2}}$$

To test which genes exhibited significantly different expression between survivors and those that succumbed to infection by *T. parva*, for each gene within the region we ran logistic

regression fitting binomial survival status against FPKM adding sex and group as covariates. This was compared to a reduced model without FPKM, by ANOVA and the likelihood ratio test.

## Prioritisation of candidate variants

Publicly available WGS data for 421 cattle [27] were processed as described above, combined with the WGS data for 43 cattle generated for this study, and variants jointly called. Additional filtering beyond that described above included extracting the target interval 15:45095457–51095457, the removal of singletons and of variants with a genotype quality (GQ) < 30, a minor allele frequency (MAF) < 0.01 or missingness ≥ 0.05. This resulted in the removal of 11 cattle and 1 variant due to missingness, and 277 variants due to low MAF. After filtering, stratified allele frequencies were calculated using Plink for the remaining 93682 variants across 453 cattle that represented different backgrounds (Boran pedigree, African taurine, African indicine, Chinese indicine, East-Asian indicine, European taurine, Indian indicine, and cattle from the Middle East). The functional consequence of variants was determined using Ensembl's Variant Effect Predictor [13] (VEP).

## Primer design and PCR amplification

Primers to genotype the candidate variant were designed using Primer3Plus [47] (https://primer3plus.com/cgi-bin/dev/primer3plus.cgi) and checked for specificity by a BLAT search against the reference genome (ARS-UCD1.2). The primers Pair1_L (5'GCTTGGGATCTGA-CAAAGGA3') and Pair1_R(5'TGGCCTCACGTTCTTCTTCT3'), synthesized at Macrogen Europe, amplified a 382bp fragment. Genomic DNA was extracted from blood samples using DNeasy Blood & Tissue Kit (Qiagen, Germany) according to the manufacturer's instructions. A 25µl PCR mix was prepared using 12.5µl OneTaq Quick-Load 2X Master Mix (New England Biolabs) with Standard Buffer, 9.5µl of nuclease-free water, 0.5µl of each 10µM primer (Pair1_L and Pair1_R) and 2µl of genomic DNA extract. The PCR was performed using an AllInOneCycler (Bioneer) with the following conditions: initial denaturation at 94˚C for 30 seconds; 30 cycles of denaturation at 94˚C, annealing at 58˚C and extension at 68˚C for 30 seconds at each step; the final extension at 68˚C for 5minutes. PCR products were sent to Macrogen Europe (Amsterdam, Netherlands) for sequencing.

## Heritability analyses

The heritability analysis was performed using ASReml (v4.2; https://asreml.kb.vsni.co.uk/) by fitting the fixed effects of sex and field trial year, with sire as a random effect. Genotype was included as a fixed effect when accounting for genetic effects. We also estimated the additive and dominance effects of each SNV. Defining AA, BB and AB to be the predicted trait values for each genotype class, p and q to be the SNV allele frequencies, the genetic effects were then calculated as follows: additive effect, a = (AA—BB)/2 and dominance effect, d = AB—[(AA + BB)/2], as outlined in Hadjipavlou et al. [48].

## Cell *in vitro* expansion and infection

The animals used for the proliferation assays were uninfected animals located on the Kapiti Research Station. The animals were genotype at the FAF1B locus and those homozygous for the reference allele (C/C) or variant allele (T/T) were selected. Peripheral blood mononuclear cells (PBMC) were isolated from venous blood and cryopreserved in liquid nitrogen. Detailed methods on the *in vitro* experiments are described in Latre de Late *et al.* [12]. Briefly, cells

were suspended in complete RPMI culture medium and infected with *T. parva* by incubation with freshly dissected salivary glands from *R. appendiculatus* fed on animals infected with *T. parva* Muguga, stabilate 3087, as previously described [49]. Tick infection levels were estimated by counting the number of infected acini in a sample of dissected salivary glands, and the sporozoite suspension adjusted to a concentration equivalent to 2000 infected acini per ml [18]. During each experiment, cells from C/C and T/T animals were infected with the same batch of sporozoites. Equal volumes of sporozoites and cells (2 x $10^7$ PBMCs) were mixed and incubated at 37˚C for 90 min with periodic mixing. Cells were centrifuged, washed, and resuspended in culture medium as described. Cells were maintained in T25 flasks with fresh culture medium added every 2 to 3 days. Live cells in cultures were quantified by trypan blue staining every two days for 12 days. To quantify live, infected cells in culture, the infrared dye LIVE/DEAD Fixable Near-IR Dead Cell Stain (ThermoFisher Scientific) was first used to identify dead cells for removal, and the remaining live cells were stained with monoclonal antibody IL-S40.2, which recognises the polymorphic immunodominant molecule (PIM) expressed on the schizont surface. Fluorescence data were acquired for 105 cells per sample using a BD FACSCanto II flow cytometer (Becton Dickinson, Belgium), and analysed using Flow Jo software (FlowJo, LLC, Oregon, USA).

## Supporting information

**S1 Table. Phenotypic data for animals involved in field trials.**
(XLSX)

**S2 Table. Daily temperature observations per animal during field trials.**
(XLSX)

**S3 Table. Results of modeling to fit survival status in response to haplotype distances.**
(XLSX)

**S4 Table. Results of eQTL modeling.**
(XLSX)

**S5 Table. Results of eQTL beta comparisons to identify response eQTL.**
(XLSX)

**S6 Table. Variant Effect Predictor output for variants identified in the genomic interval associated with tolerance to *Theileria parva* infection, 15:45–51 Mb.**
(XLSX)

**S7 Table. Data underlying Fig 5.**
(XLSX)

## Acknowledgments

The authors are very grateful to the management of the Ol Pejeta Conservancy, Private Bag, Nanyuki 10400, Kenya for their assistance in performing the field studies. The authors would also like to thank Mingyan Yu and Cynthia Onzere of ILRI, for preparing the DNA samples for the genomic analyses.

## Author Contributions

**Conceptualization:** David Wragg, Elizabeth A. J. Cook, Tatjana Sitt, Johanneke D. Hemmink, Maurine C. Chepkwony, Regina Njeru, Jessica Powell, Antoinette A. Miyunga, Gideon

Ndambuki, Stephen Mwaura, Harriet Auty, Musa Hassan, Karen Marshall, Timothy Connelley, Liam J. Morrison, B. Mark deC. Bronsvoort, W. Ivan Morrison, Philip G. Toye, James G. D. Prendergast.

**Data curation:** David Wragg, Rebecca Callaby, Philip G. Toye, James G. D. Prendergast.

**Formal analysis:** David Wragg, Elizabeth A. J. Cook, Perle Latré de Laté, Tatjana Sitt, Johanneke D. Hemmink, Maurine C. Chepkwony, E. Jane Poole, Jessica Powell, Edith A. Paxton, Antoinette A. Miyunga, Oswald Matika, Philip G. Toye, James G. D. Prendergast.

**Funding acquisition:** W. Ivan Morrison, Philip G. Toye, James G. D. Prendergast.

**Investigation:** David Wragg, Elizabeth A. J. Cook, Tatjana Sitt, Johanneke D. Hemmink, Maurine C. Chepkwony, Regina Njeru, Jessica Powell, Antoinette A. Miyunga, Gideon Ndambuki, Stephen Mwaura, Karen Marshall, Timothy Connelley, Liam J. Morrison, B. Mark deC. Bronsvoort, Philip G. Toye, James G. D. Prendergast.

**Methodology:** David Wragg, Elizabeth A. J. Cook, Perle Latré de Laté, Tatjana Sitt, Johanneke D. Hemmink, Maurine C. Chepkwony, Regina Njeru, E. Jane Poole, Jessica Powell, Antoinette A. Miyunga, Gideon Ndambuki, Stephen Mwaura, Oswald Matika, Musa Hassan, Karen Marshall, Timothy Connelley, Liam J. Morrison, B. Mark deC. Bronsvoort, Philip G. Toye, James G. D. Prendergast.

**Project administration:** Philip G. Toye, James G. D. Prendergast.

**Resources:** Rebecca Callaby, Andrea Talenti, Philip G. Toye, James G. D. Prendergast.

**Software:** David Wragg, Andrea Talenti.

**Supervision:** Philip G. Toye, James G. D. Prendergast.

**Visualization:** David Wragg, James G. D. Prendergast.

**Writing – original draft:** David Wragg, Elizabeth A. J. Cook, James G. D. Prendergast.

**Writing – review & editing:** David Wragg, Elizabeth A. J. Cook, Perle Latré de Laté, Tatjana Sitt, Johanneke D. Hemmink, Maurine C. Chepkwony, Regina Njeru, Jessica Powell, Edith A. Paxton, Rebecca Callaby, Andrea Talenti, Antoinette A. Miyunga, Gideon Ndambuki, Stephen Mwaura, Harriet Auty, Oswald Matika, Musa Hassan, Karen Marshall, Timothy Connelley, Liam J. Morrison, B. Mark deC. Bronsvoort, W. Ivan Morrison, Philip G. Toye, James G. D. Prendergast.

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
