## [Decision Letter · Decision Letter 0]

11 Nov 2021

Dear Dr Wragg,

Thank you very much for submitting your Research Article entitled 'A locus conferring tolerance to Theileria infection in African cattle' to PLOS Genetics.

The manuscript was fully evaluated at the editorial level and by independent peer reviewers. We apologize for the long duration of the review process. The reviewers appreciated the attention to an important topic but identified some concerns that we ask you address in a revised manuscript.

We therefore ask you to modify the manuscript according to the review recommendations. Your revisions should address the specific points made by each reviewer.

[LINK]

Yours sincerely,

Tosso Leeb

Associate Editor

PLOS Genetics

Scott Williams

Section Editor: Natural Variation

PLOS Genetics

In addition to the comments by the 2 reviewers, please also adress this point:

In your filtering pipeline, you only considered variants with "high impact" according to VEP prediction. This seems overly stringent to me. I understand the rationale of prioritizing protein coding variants. However, there are arguably much more missense variants with detrimental effects on protein function than "high impact" variants in the genome. Please expand your analysis to include "moderate impact" variants. If you include the "moderate impact" variants, will this lead to more than 4 candidate causal variants? Please provide Table S6 in a format that allows easy filtering and sorting of the variants (e.g. as an Excel file).

Reviewer's Responses to Questions

**Comments to the Authors:**

Reviewer #1: The presented work is the first to report a potential causative gene variant underlying the heritable tolerance to ECF disease that has been known for some time among local African Zebu cattle. Given the impact of this infectious disease on animal welfare and the economy, this is a major breakthrough. The specific outcome of this study revealed a nonsense variant in the bovine FAF1 gene that is confirmed to be significantly associated with survival outcome after experimental infection. Therefore, a major gene for natural (host) resistance to infection with theileria has been identified.

The functional effect of the loss-of-function variant remains unclear. Possible Faf1 knockout mouse model are not mentioned. Obviously Faf1 plays a key role in the dopaminergic neurodegeneration according to DOI: 10.1093/hmg/ddt006. It appears to be a Mendelian trait as TT homozygotes are at an advantage but there is also an effect in heterozygous carriers: which type of inheritance best explains this? In general susceptibility/resistance against infectious diseases represents a complex trait of polygenic nature.

Therefore, the question arises as to what proportion of the total variance this specific variant explains. Whether this finding provides full protection against natural infection is not completely clear. The option mentioned in the conclusion to spread this beneficial allele by selection or even targeted genome editing seems questionable to me, as this requires a targeted breeding program and the use of artificial insemination of appropriate bulls with the desired FAF1 genotype.

Specific comments:

The indicine nature of the Boran breed could be mentioned once at the beginning.

The nonsense variant in FAF1 should be mentioned more clearly, at least in the summary and maybe even in the title.

The International Society for Animal Genetics (ISAG) desires the use of human gene and variant nomenclature rules for domestic animals. The gene names should adhere to HGNC (www.genenames.org/). According to HGVS recommendations, the term “polymorphism” should be exchanged with “variant” (see https://varnomen.hgvs.org/).

Please adhere to the published guidelines of sequence variant nomenclature (http://varnomen.hgvs.org/) e.g. the GenBank accessions of the used mRNA and protein entries should be mentioned to understand the given positions.

The discussion turns out to be relatively short, conspicuously little literature is cited. For example a cross comparison to the very few known similar major gene effects against infectious diseases in domestic animals, e.g. TMEM154-related SRLV-susceptibility in sheep could enrich the discussion.

Figure 1 might be moved into the supplement as it shows very detailed phenotypic records but lacking the individual FAF1 genotypes.

The very recent run9 variant catalogue of the 1000 Bulls genome Project includes also many indicine cattle genomes, maybe this resource could be consulted to improve the presentation of the allele distribution within taurine and indicine cattle.

I wonder if there were attempts to look for structural variants in the sequenced genomes.

Reviewer #2: In this interesting and important study, a genetic analysis of cattle which demonstrate heritable tolerance to infection by T. parva (the cause of East Coast Fever, ECF) was performed. Evidence has accumulated over many years to suggest that the frequency of natural tolerance alleles differs between cattle from endemic and non-endemic areas. The current study is the first to map genetic loci driving tolerance to ECF. Current treatment and prevention measures to control ECF are imperfect and expensive. The discovery of a genetic marker to aid selectively breeding tolerant cattle has huge economic significance. 4 genes associated with the variants conferring tolerance; two olfactory receptors (OR51H1 and OR51G1), one paralog of haemaglobin subunit beta (HBB) and one paralog of FAS-associated factor 1 (FAF1). Because T. parva infected cells are resistant to FAS-induced cell death, FAF1 was considered for further analysis. To test the effect of tolerant (T/T) vs reference (C/C) FAF1B alleles on in vitro expansion of T. parva infected cells, PBMCs were isolated from 12 cattle with the FAF1B T/T (tolerant) and 12 with the C/C (susceptible) genotype. For the 12 T/T cattle, 6 were from tolerant pedigree, and 6 were from unrelated cattle. Cells from the C/C genotype exhibited significantly increased growth rate (as counted by manual counting of cells stained with trypan blue) and parasite infection (as measured by staining the parasite with fluorescently labelled antibodies against the parasite surface, PIM).

As the editor knows, I am not an expert in genetics, and cannot critically review the genetic analysis tools applied. I therefore focus my review on the figure 5 and the overall impact. I do however also have a few small comments and questions about the genetic linkage part of the study.

Table 1: what is the explanation for the rather high proportion of cattle deaths in sire 3167 offspring 2018 study (both 2017 and 2018 studies involved 2nd generation offspring). Why are the 2017 and 2018 studies so different?

Line 139: could you explain a bit more clearly why the 29 animals from the 2018 study were selected (where only 30% of sire 3167 survived).

A genomic region significantly associated with survival outcome was identified. 58066 variants, and 235 genes in the region were identified. 55 variants were selected for study. Can you explain what criteria were used to select the 55 variants predicted to have a functional consequence? (line 153). The authors state in the method that Ensembl’s variant effect predictor was used. As a non-geneticist I wondered what this means.

Figure 5: This experiment nicely supported the observation made that PBMCs isolated from cattle that were homozygous for the “reference” FAF1B SNP (C/C) were more highly infected with T. parva (as shown by staining with the schizont surface marker PIM), and correspondingly exhibited increased cell proliferation. Conversely cells isolated from animals that were homozygous for the “tolerant” SNP (T/T) were less susceptible to infection and consequently underwent less proliferation. These experiments were well controlled by using same number of PBMCs and normalizing the number of sporozoites obtained from the ticks by counting infected acini. Importantly cells from C/C and T/T cows were infected with the same batch of sporozoites, meaning that variations in sporozoite infectivity is less likely to impact the data.

It would be great to see some further functional validation of the potential role of FAF1B mutations in driving the tolerance phenotype. The authors already point out that FAF1B expression could not be detected in blood either before or after infection (pre-empting an obvious question and making it more difficult to imagine that FAF1B in fact does play an important role). It would indeed (as the authors already state) be important to check for FAF1B expression in other tissues.

I can think of a number of experiments that would be interesting to perform. One suggestion is to test the susceptibility of FAF1B C/C vs T/T cells before and after infection to Fas mediated apoptosis. A second experiment worth considering is to use CRISPR/Cas9 to introduce the FAF1B “tolerant” SNP (or otherwise knock out or knock down FAF1B) in cells isolated from susceptible cattle and then test their susceptibility to infection. This would be a great way to test whether it really is the SNP in FAF1B that drives tolerance.

However, I do very much appreciate the precise way the authors present their work, they do not attempt to over-sell or over-state their findings (e.g. line 312: “….. further worked required to elucidate any causal relationship between the FAF1 paralogue and enhanced tolerance…….”, and as such I would not insist on any additional experiments.

**Have all data underlying the figures and results presented in the manuscript been provided?**

Reviewer #1: Yes

Reviewer #2: Yes

PLOS authors have the option to publish the peer review history of their article (what does this mean?). If published, this will include your full peer review and any attached files.

Reviewer #1: No

Reviewer #2: No

---

## [Decision Letter · Decision Letter 1]

14 Feb 2022

Dear Dr Wragg,

We are pleased to inform you that your manuscript entitled "A locus conferring tolerance to Theileria infection in African cattle" has been editorially accepted for publication in PLOS Genetics. Congratulations!

There is one minimal issue that can be taken care of during the next steps: In your manuscript, you interchangeably use the terms "stop gained", "stop gain" and "nonsense" variant. Please consider to use a consistent terminology for this type of variant.

Yours sincerely,

Tosso Leeb

Associate Editor

PLOS Genetics

Scott Williams

Section Editor: Natural Variation

PLOS Genetics

Comments from the reviewers (if applicable):

Reviewer's Responses to Questions

**Comments to the Authors:**

Reviewer #2: I am satisfied with the author's response to my comments.

**Have all data underlying the figures and results presented in the manuscript been provided?**

Reviewer #2: Yes

PLOS authors have the option to publish the peer review history of their article (what does this mean?). If published, this will include your full peer review and any attached files.

Reviewer #2: No

**Data Deposition**

http://datadryad.org/submit?journalID=pgenetics&manu=PGENETICS-D-21-01288R1

**Press Queries**

---

## [Editor Report · Acceptance letter]

30 Mar 2022

PGENETICS-D-21-01288R1 

A locus conferring tolerance to *Theileria* infection in African cattle 

Dear Dr Prendergast, 

We are pleased to inform you that your manuscript entitled "A locus conferring tolerance to *Theileria* infection in African cattle" has been formally accepted for publication in PLOS Genetics! Your manuscript is now with our production department and you will be notified of the publication date in due course.

With kind regards,

Livia Horvath

PLOS Genetics

On behalf of:
